

# Serum microRNAs as potential new biomarkers for cisplatin resistance in gastric cancer patients

Lei Jin[1,2,*], Nan Zhang[2,3,*], Qian Zhang[2,4], Guoqian Ding[1,2], Zhenghan Yang[2,3] and Zhongtao Zhang[1,2]

[1] Department of General Surgery, Beijing Friendship Hospital, Capital Medical University, Beijing, China
[2] National Clinical Research Center for Digestive Diseases, Beijing, China
[3] Department of Radiology, Beijing Friendship Hospital, Capital Medical University, Beijing, China
[4] Clinical Epidemiology and EBM Center, Beijing Friendship Hospital, Capital Medical University, Beijing, China
[*] These authors contributed equally to this work.

Corresponding authors
Zhenghan Yang,
yangzhenghan@ccmu.edu.cn
Zhongtao Zhang,
zhangzhongtao@ccmu.edu.cn

## ABSTRACT

**Background**. microRNAs (miRNAs) have been studied for their role in the early detection of several diseases. However, there is no current information on the systematic screening of serum-derived cisplatin resistance biomarkers in gastric cancer (GC).
**Methods**. Cisplatin-resistant GC cell lines were screened for dysregulated miRNAs using small RNA sequencing (sRNA-seq) and miRNAs were functionally annotated using bioinformatics analyses. Real-time quantitative polymerase chain reaction (RT-qPCR) was used to validate the miRNA-relative transcription levels in GC cells and in 74 GC patients. We analyzed the associations between the clinical characteristics of the patients and their miRNA expression. Receiver operating characteristic (ROC) analysis was used to evaluate the diagnostic value for serum-derived cisplatin resistance.
**Results**. Seven miRNAs were identified from 35 differentially expressed miRNAs between the MGC803/DDP and MGC803 cells in a public database. We found four miRNA candidates (miR-9-3p, miR-9-5p, miR-146a-5p, and miR-433-3p) that were significantly associated with chemotherapy responses in GC cells and patients. miR-9-5p (AUC = 0.856, 95% CI [0.773–0.939], $p < 0.0001$) and a combined group (miR-9-5p + miR-9-3p + miR-433-3p) (AUC = 0.915, 95% CI [0.856–0.975], $P < 0.0001$) distinguished chemoresistant GC patients from chemosensitive GC patients.
**Conclusions**. Our study reveals the potential therapeutic use of two serum-based biomarkers, miR-9-5p and a combined group (miR-9-5p + miR-9-3p + miR-433-3p), as indicators for the successful use of cisplatin in GC patients.

## INTRODUCTION

Gastric cancer is the fourth most commonly diagnosed malignant cancer and the second leading cause of tumor-related deaths worldwide (*Siegel, Miller & Jemal, 2019*). Surgery combined with chemotherapy is an emerging and treatment for GC.

Cisplatin (DDP) is recognized as a first-line chemotherapy drug for GC patients in a progressive stage of the disease. The current long-term survival rate of GC patients is poor due to the high prevalence of drug-resistance, metastasis, and recurrence (*Zong et al., 2016*). DDP resistance in most patients is inevitable and results in failed treatment, while the long-term use and repeated administration of DDP leads to severe side effects (*Kovalchuk et al., 2008*). The development of novel biomarkers for chemotherapy-resistant GC is critical for improving the prognostic efficacy in patients.

miRNAs are a discovered class of small noncoding RNAs containing 19–25 nucleotides (*Bartel, 2004*). miRNAs have been identified in many biofluids, including serum, plasma, and urine (*Etheridge et al., 2011*), suggesting that circulating miRNAs could be used as minimally invasive biomarkers for cancer and other diseases (*Etheridge et al., 2011*; *Vychytilova-Faltejskova et al., 2016*; *Schultz et al., 2014*). The use of biomarkers is noninvasive, more comfortable than endoscopic examination, and may reflect the heterogeneity of the disease. Dysregulation of miRNAs may lead to DDP resistance in many tumors (*Sorrentino et al., 2008*; *Yu et al., 2010*), but the role of miRNAs in the chemo-response of GC is not fully understood.

miRNAs in human tumor serum specimens may act as markers to predict the treatment sensitivity and prognosis in many cancer types (*Ueda et al., 2010*; *Goswami et al., 2013*; *Hur et al., 2015*). Emerging evidence reveals that specific miRNAs in serum specimens may aid in the early diagnosis of malignancies (*Masuda & Izpisua Belmonte, 2014*; *Lin et al., 2015*) and in the determination of the survival prognosis for cancer patients after surgery (*Hu et al., 2010*). However, it is not understood how specific miRNAs in serum might predict a chemotherapy response in GC patients.

Small RNA-sequencing was used to analyze the microRNA profiles in GC cells to find the dysregulated miRNAs. RT-qPCR and correlation analyses were used to validate the results of sRNA-seq, which was used to select the miRNA candidates. We also attempted to develop specific biomarkers to determine the chemotherapy response of GC.

## MATERIALS & METHODS

### Cell culture

The MGC803 cell line was obtained from the Chinese Academy of Medical Sciences (Beijing, China). DDP-resistant GC cells were established by culturing MGC803 cells in a continuous stepwise fashion with gradually increasing concentrations of DDP from 0.05 to 5 µg/ml over a period of 15 months. This resulted in a shift towards cell proliferation and an apoptosis phenotype (*Hong et al., 1988*; *Yu, Ma & Chang, 2000*). The cell line was continuously cultured with 2 µg/ml DDP to maintain its resistance. These two cell lines were cultured following standard culturing procedures and identified by short tandem repeat (STR) profile data compared with the ATCC, DSMZ, or JCRB databases (Cobioer Biosciences Co., Ltd, Los Altos, CA, USA).

### Patient enrollment and ethics statement

Serum specimens were collected from 74 GC patients who met the following criteria: (1) histologically confirmed gastric adenocarcinoma; (2) DDP-based chemotherapy used as

first-line treatment; (3) received at least 2 cycles of neoadjuvant chemotherapy or palliative treatment; (4) chemotherapy efficacy could be evaluated by unenhanced and enhanced computed tomography (CT) after 2 or 3 cycles; (5) serum samples were collected before the first chemotherapy. All patients provided informed consent prior to the collection of a blood sample. This study was approved by the Ethics Committee (Approval Number: 2018-P2-045-01) of Beijing Friendship Hospital, Capital Medical University and met the ethical requirements of the Declaration of Helsinki. Prechemotherapy serum samples were collected between January 2015 and January 2019 from patients who received neoadjuvant chemotherapy or palliative treatment. The serum samples were collected and processed following the standard operating procedure of the Early Detection Research Network. The chemotherapy principle was executed according to the guidelines of the American Society of Clinical Oncology (ASCO). The chemotherapy response effect was evaluated according to CHOI's principle (*Choi et al., 2004*). Treatment resistance was evaluated by the existing progressive disease (PD) in <3 treatment cycles, and the treatment response was evaluated by the existing complete response (CR), stable disease (SD), or partial response (PR) lasting for 2 or 3 cycles. The chemotherapy response-sensitive group included CR and PR patients, whereas the chemotherapy response-resistant group included SD and PD patients.

## RNA extraction

Total RNA was isolated from the GC cells using Trizol (Invitrogen, USA). The same amount of Caenorhabditis elegans cel-39-3p miRNA was spiked into each serum sample as an external calibration for RNA extraction, reverse transcription, and miRNA amplification. Total RNA was extracted and purified from serum using the miRNeasy Serum/Plasma kit (Qiagen, cat. 217184). The quantity and integrity of the RNA yield was assessed using the Qubit2.0 fluorometer (Life Technologies, USA) and Agilent 2200 TapeStation (Agilent Technologies, USA), separately.

## sRNA-seq and data analysis

Total RNA (1 μg) of GC cells were used to prepare small RNA libraries by NEBNext Multiplex Small RNA Library Prep Set for Illumina (NEB, USA) according to the manufacturer's instructions. The libraries were sequenced by HiSeq 2500 (Illumina, USA) with single-end 50 bp reads at RiboBio Co. Ltd (RiboBio, China). Raw reads were processed by FastQC to get clean reads by filtering out those containing adapter, poly 'N', were of low quality, or were smaller than 17nt reads. Mapping reads were obtained by mapping clean reads to the reference genome from the BWA software. miRDeep2 was used to identify known mature miRNA based on miRBase21 (http://www.miRBase.org) and to predict novel miRNA. The databases of Rfam12.1 (http://www.rfam.xfam.org) and piRNABank (http://www.pirnabank.ibab.ac.in) were used to identify ribosomal RNA (rRNA), transfer RNA (tRNA), small nuclear RNA (snRNA), small nucleolar RNA (snoRNA), and piwi-interacting RNA (piRNA) by BLAST. miRNA expressions were calculated in RPM (reads per million) values (RPM = (number of reads mapping to miRNA/number of reads in clean data) $\times 10^6$). The expression levels were normalized by RPM [(number of reads mapping to miRNA/number of reads in clean data) $\times 10^6$].
## Bioinformatics analysis

The differential expression between the two sample sets was calculated using the edgeR algorithm according to the criteria of |log2 (Fold Change) | $\geq 1$ and $P$-value $<0.05$. TargetScan, miRDB, miRTarBase, and miRWalk were used to predict the target genes of selected miRNA. Patients with different expression levels of miRNAs were determined to have shorter or longer overall survival (OS) times for 10-year OS using the Kaplan–Meier survival curve data according to the Kaplan–Meier Plotter database (http://kmplot.com) (*Szász et al., 2016*; *Nagy et al., 2018*), the OncoLnc database (http://www.oncolnc.org/) (*Anaya, 2016*), and the OncomiR database (http://www.oncomir.org/) (*Wong et al., 2018*).

## RT-qPCR

The primers for miRNA detection were purchased from FulenGene Company (Guangzhou, China) and their sequence is listed in Table 1. Total RNA was reverse transcribed using the All-in-One$^{TM}$ miRNA RT-qPCR Kit (GeneCopoeia, USA) for miRNA analysis. The All-in-One$^{TM}$ miRNA RT-qPCR Detection Kit (GeneCopoeia, USA) was used to measure miRNAs quantitatively. All samples were normalized by the initial biofluid input volume used for RNA extraction and were calibrated by the spike-in cel-39-3p to eliminate the minute bias caused by different RNA isolation efficiencies and PCR efficiencies among samples. U6 or let-7g-5p was used as an endogenous control to normalize the relative number of miRNAs. ABI 7500 real-time fast PCR system (Applied Biosystems, USA) was used to achieve the relative quantitation of miRNA expression and the data was analyzed using the affiliated software. The Ct attenuation value of each type of miRNA in each of the serum samples was corrected by the internal reference cel-39-3p and the two housekeeping genes, U6 and let-7g-5p. The $2^{-\Delta\Delta}$ method was used to calculate the relative expression of each miRNA. Each sample per reaction was performed in triplicate.

## Statistical analysis

miRNA expression levels in the serum samples were divided into high and low groups taking the median as the cutoff value by RT-qPCR. Statistical differences for *in vitro* experiments were analyzed using student's unpaired *t* tests. Associations between the clinical parameters of the patients and their miRNA expression were analyzed using the Mann–Whitney test. The AUC was used to assess the diagnostic accuracy of the predictors. Logistic regression was used to develop a panel of combined biomarkers to predict the probability of GC chemotherapy response. All data were expressed as mean $\pm$ SD. Each experiment was repeated independently at least three times. Quantitative data were analyzed and graphed using SPSS 23.0, MedCalc, and GraphPad Prism 7. Differences were considered to be significant at ****$P < 0.0001$, ***$P < 0.001$, **$P < 0.01$, and *$P < 0.05$.

## RESULTS

### Analysis of sRNA-seq data

The results of sRNA-seq in MGC803/DDP cells (Fig. 1A) and MGC803 cells (Fig. 1B) showed that the sRNA sequence included miRNA, tRNA, rRNA, snRNA, snoRNA, piRNA,

**Table 1  Primers of RT-qPCR.**

| miRNA symbol | Article number | Sequence |
|---|---|---|
| miR-9-3p | HmiRQP0824 | CCATAAAGCTAGATAACCGAAAGTAA |
| miR-9-5p | HmiRQP0825 | TCTTTGGTTATCTAGCTGTATGAAA |
| miR-146a-5p | HmiRQP0196 | AGAACTGAATTCCATGGGTTAA |
| miR-370-3p | HmiRQP0456 | GCTGGGGTGGAACCTGGTAA |
| miR-433-3p | HmiRQP0502 | TGATGGGCTCCTCGGTGTAA |
| miR-519a-5p | HmiRQP0591 | TAGAGGGAAGCGCTTTCTGAAA |
| miR-522-5p | HmiRQP0591 | TAGAGGGAAGCGCTTTCTGAAA |
| let-7g-5p | HmiRQP0015 | TGAGGTAGTAGTTTGTACAGTTAA |
| U6 | HmiRQP9001 | Not available |

and other RNA. Thirty-five miRNAs were selected and intersected, of which 11 miRNAs were up-regulated, and 24 down-regulated at higher levels in MGC803/DDP cells than the MGC803 cells (Fig. 1C). These 35 differentially expressed miRNAs in sRNA-seq data are shown in Table 2.

## Bioinformatics analysis of selected miRNA

A large cohort analysis was conducted using Kaplan–Meier survival data according to the Kaplan–Meier Plotter database in order to determine the role of the 35 differentially expressed miRNAs as a potential prognostic factor. We observed that a total of seven differentially expressed miRNAs showed statistical differences ($p < 0.05$) for 10-year overall survival (OS) (Table 2), which indicated that these miRNAs could be used as valid biomarkers for chemotherapy response and overall survival. The differentially expressed miRNAs observed were miR-9 (including miR-9-3p and miR-9-5p, Fig. 2A), miR-146a (Fig. 2B), miR-370 (Fig. 2C), miR-433 (Fig. 2D), miR-519a (Fig. 2E), and miR-522 (Fig. 2F). TargetScan, miRDB, miRTarBase, and miRWalk were used to predict the target genes of the seven selected miRNAs, including miR-9-3p (Fig. 3A), miR-9-5p (Fig. 3B), miR-146a-5p (Fig. 3C), miR-370-3p (Fig. 3D), miR-433-3p (Fig. 3E), miR-519a-5p (Fig. 3F), and miR-522-5p (Fig. 3G). Supplementary material from four different databases (Table S1) was added to the common target genes of the 7 miRNAs (Fig. 3). These 7 miRNAs were selected as miRNA candidates.

## Validation of sRNA-seq data by RT-qPCR analysis

GC cell results showed that the levels of miR-146a-5p, miR-519a-5p, and miR-522-5p were significantly downregulated in MGC803/DDP cells. The levels of miR-9-3p, miR-9-5p, miR-370-3p, and miR-433-3p were significantly upregulated in MGC803/DDP cells (Table 3, Fig. 4A). miR-519a-5p and miR-522-5p changed consistently with sRNA-seq, but the opposite results were seen from the other five miRNAs. miR-519a-5p and miR-522-5p had the same primer and exhibited the same result.

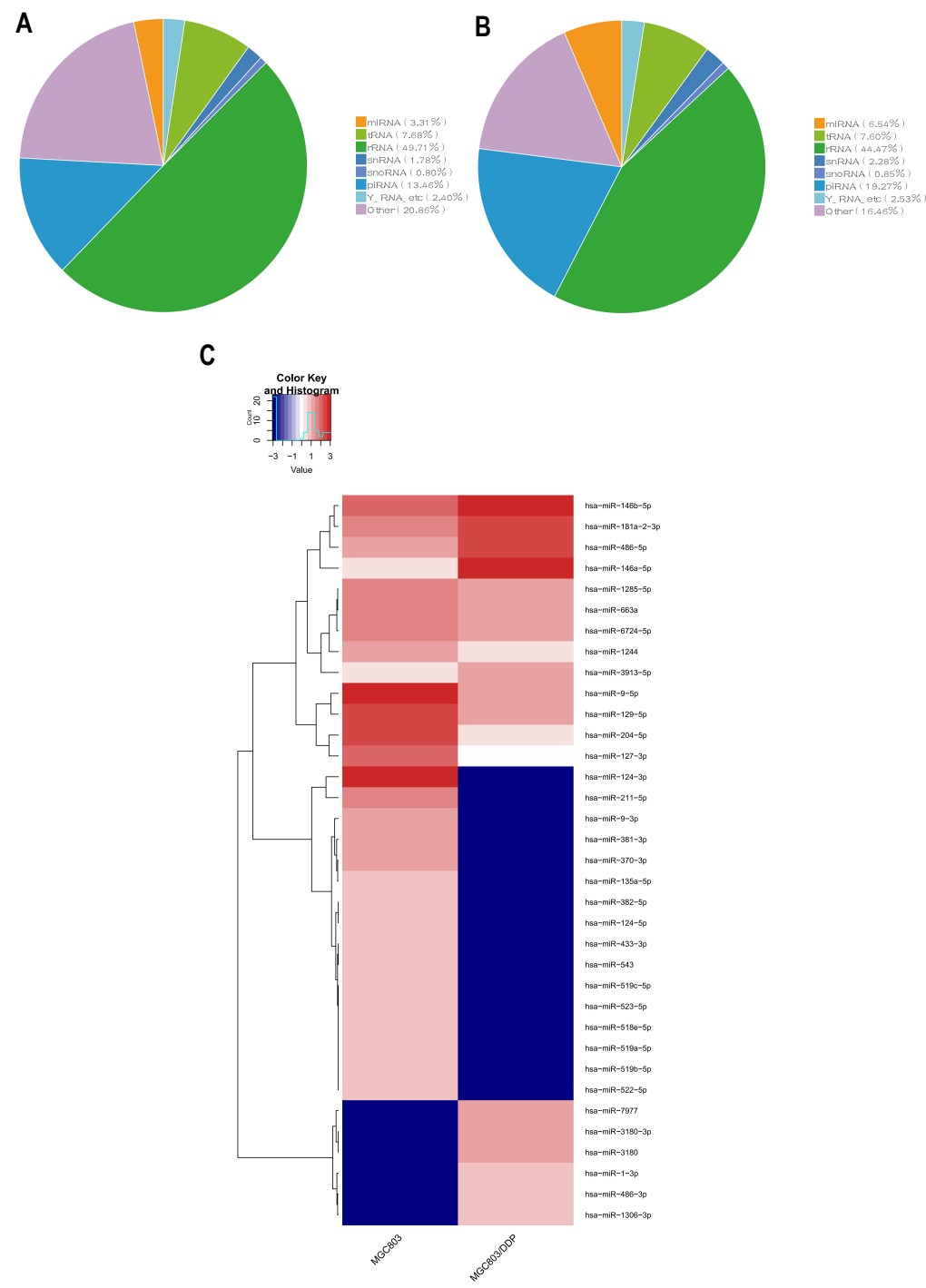

**Figure 1  Analysis for sRNA-seq data of MGC803/DDP and MGC803 groups.** (A, B) Pie charts showing the percentage of different kinds of sRNA to the mapped reads for MGC803 group (A) and MGC803/DDP group (B). (C) A heat map of the 35 diferentially expressed miRNAs in MGC803/DDP and MGC803 groups was showed.

**Table 2  35 differentially expressed miRNAs in sRNA-seq data and Kaplan–Meier Plotter analysis.**

| miRNA | Counts | | log$_2$FC | *P*-value | Kaplan–Meier Plotter | |
|---|---|---|---|---|---|---|
| | MGC803 | MGC803/DDP | | | Hazard Ratio | *P*-value |
| hsa-miR-124-3p | 457 | 0 | −12.2235 | 1.2576E-16 | 0.75 (0.55 ∼1.02) | 0.0610 |
| hsa-miR-9-5p | 1155 | 12 | −7.2625 | 1.0889E-15 | 1.56 (1.14 ∼2.13) | 0.0047** |
| hsa-miR-204-5p | 238 | 3 | −6.9301 | 4.0321E-12 | 0.73 (0.50 ∼1.06) | 0.0930 |
| hsa-miR-146a-5p | 4 | 486 | 6.1991 | 4.1414E-11 | 0.70 (0.52 ∼0.95) | 0.0230* |
| hsa-miR-129-5p | 352 | 14 | −5.3287 | 1.9485E-10 | 1.39 (0.99 ∼1.93) | 0.0540 |
| hsa-miR-127-3p | 84 | 1 | −6.8789 | 1.1322E-09 | 1.30 (0.95 ∼1.79) | 0.0990 |
| hsa-miR-211-5p | 62 | 0 | −9.3436 | 1.4691E-08 | 0.74 (0.53 ∼1.04) | 0.0810 |
| hsa-miR-486-5p | 21 | 298 | 3.1290 | 3.7724E-05 | 0.77 (0.55 ∼1.07) | 0.1200 |
| hsa-miR-9-3p | 18 | 0 | −7.5647 | 0.0002 | 1.56 (1.14 ∼2.13) | 0.0047** |
| hsa-miR-381-3p | 13 | 0 | −7.0982 | 0.0010 | 1.26 (0.90 ∼1.75) | 0.1700 |
| hsa-miR-3180 | 0 | 20 | 7.0281 | 0.0010 | 0.76 (0.55 ∼1.05) | 0.0910 |
| hsa-miR-3180-3p | 0 | 20 | 7.0281 | 0.0010 | 0.76 (0.55 ∼1.05) | 0.0910 |
| hsa-miR-370-3p | 11 | 0 | −6.8591 | 0.0029 | 1.56 (1.14 ∼2.13) | 0.0053** |
| hsa-miR-135a-5p | 10 | 0 | −6.7228 | 0.0042 | 1.16 (0.85 ∼1.58) | 0.3500 |
| hsa-miR-146b-5p | 78 | 481 | 1.9311 | 0.0058 | 1.16 (0.83 ∼1.62) | 0.3900 |
| hsa-miR-522-5p | 9 | 0 | −6.5723 | 0.0061 | 1.38 (1.01 ∼1.89) | 0.0400* |
| hsa-miR-519b-5p | 9 | 0 | −6.5723 | 0.0061 | 0.79 (0.58 ∼1.07) | 0.1200 |
| hsa-miR-519a-5p | 9 | 0 | −6.5723 | 0.0061 | 1.39 (1.02 ∼1.91) | 0.0360* |
| hsa-miR-518e-5p | 9 | 0 | −6.5723 | 0.0061 | 0.78 (0.57 ∼1.06) | 0.1200 |
| hsa-miR-523-5p | 9 | 0 | −6.5723 | 0.0061 | 0.77 (0.57 ∼1.05) | 0.0930 |
| hsa-miR-519c-5p | 9 | 0 | −6.5723 | 0.0061 | 0.79 (0.58 ∼1.08) | 0.1400 |
| hsa-miR-7977 | 0 | 14 | 6.5182 | 0.0061 | 0.75 (0.54 ∼1.05) | 0.0900 |
| hsa-miR-543 | 8 | 0 | −6.4043 | 0.0092 | 0.76 (0.55 ∼1.05) | 0.0990 |
| hsa-miR-433-3p | 8 | 0 | −6.4043 | 0.0092 | 1.62 (1.20 ∼2.20) | 0.0017*** |
| hsa-miR-1244 | 16 | 4 | −2.6460 | 0.0144 | 0.74 (0.53 ∼1.02) | 0.0630 |
| hsa-miR-486-3p | 0 | 10 | 6.0391 | 0.0219 | 0.77 (0.55 ∼1.07) | 0.1200 |
| hsa-miR-6724-5p | 28 | 12 | −1.9009 | 0.0264 | 0.74 (0.53 ∼1.02) | 0.0630 |
| hsa-miR-663a | 32 | 14 | −1.8732 | 0.0287 | 0.74 (0.53 ∼1.02) | 0.0630 |
| hsa-miR-124-5p | 6 | 0 | −5.9949 | 0.0348 | 0.75 (0.55 ∼1.02) | 0.0610 |
| hsa-miR-382-5p | 6 | 0 | −5.9949 | 0.0348 | 1.31 (0.95 ∼1.81) | 0.1000 |
| hsa-miR-1306-3p | 0 | 9 | 5.8895 | 0.0348 | 0.82 (0.60 ∼1.13) | 0.2300 |
| hsa-miR-1-3p | 0 | 8 | 5.7226 | 0.0348 | 1.30 (0.95 ∼1.76) | 0.0980 |
| hsa-miR-1285-5p | 30 | 15 | −1.6819 | 0.0366 | 0.74 (0.53 ∼1.02) | 0.0630 |
| hsa-miR-3913-5p | 2 | 20 | 2.5736 | 0.0382 | 1.23 (0.88 ∼1.72) | 0.2300 |
| hsa-miR-181a-2-3p | 60 | 260 | 1.4220 | 0.0429 | 1.35 (0.99 ∼1.84) | 0.0530 |

## Expression levels and functions of miRNAs in human GC clinical specimens

RT-qPCR tests were run on the serum samples from 74 GC patients to identify the potential miRNAs. 2 housekeeping genes (U6 and let-7g-5p) and a spike-in cel-39-3p served as the internal references.

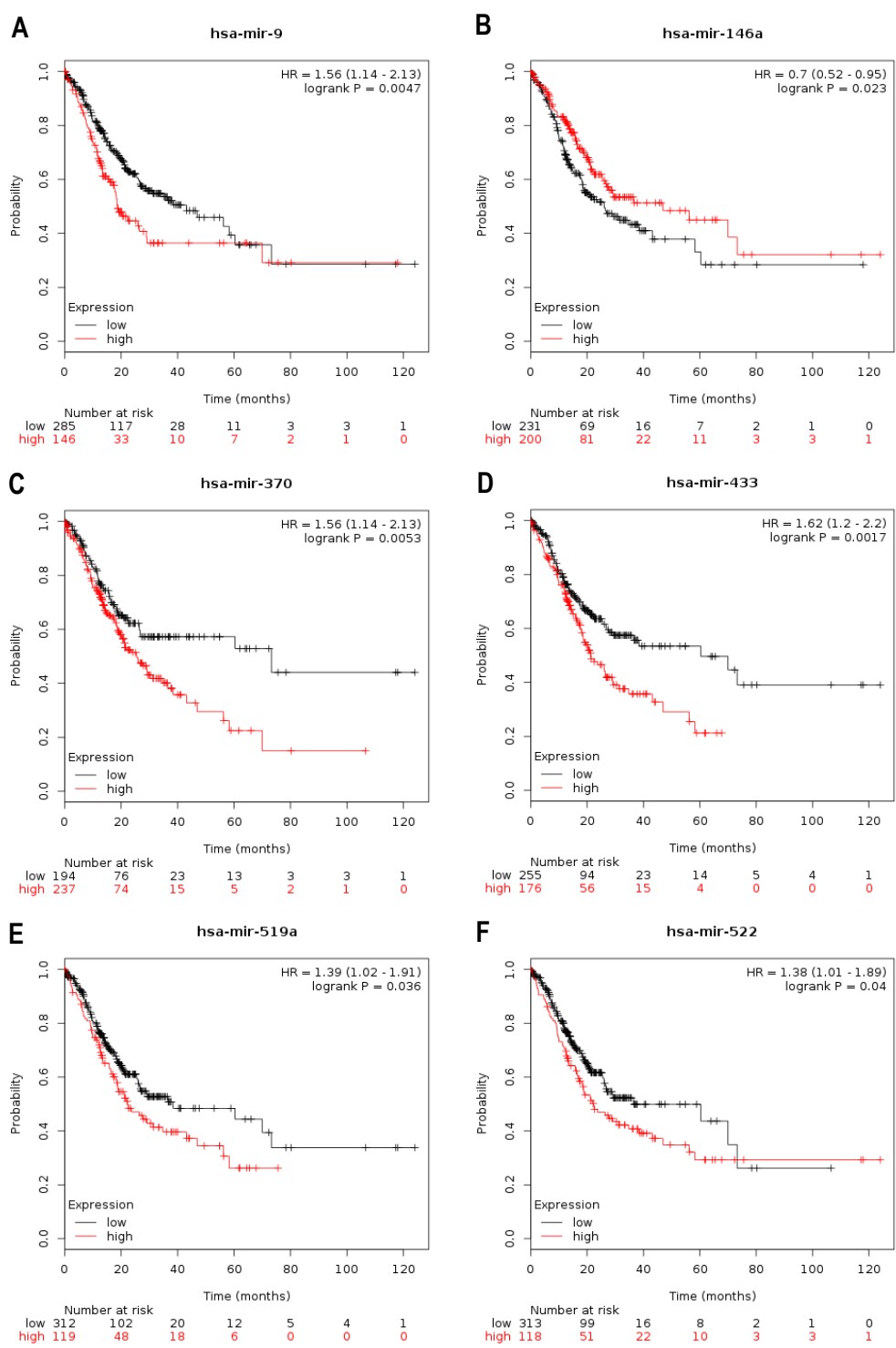

**Figure 2** **The role of 7 selected miRNAs was ascertained using Kaplan–Meier survival data according to Kaplan–Meier Plotter.** (A–F) Kaplan-Meier survival curves suggested that patients with high miR-9 (A), miR-370 (C), miR-433 (D), miR-519a (E) and miR-522 (F) levels had lower OS times for 10-year OS than those with low levels, and patients with high miR-146a (B) levels had higher OS times for 10-year OS than those with those low levels.

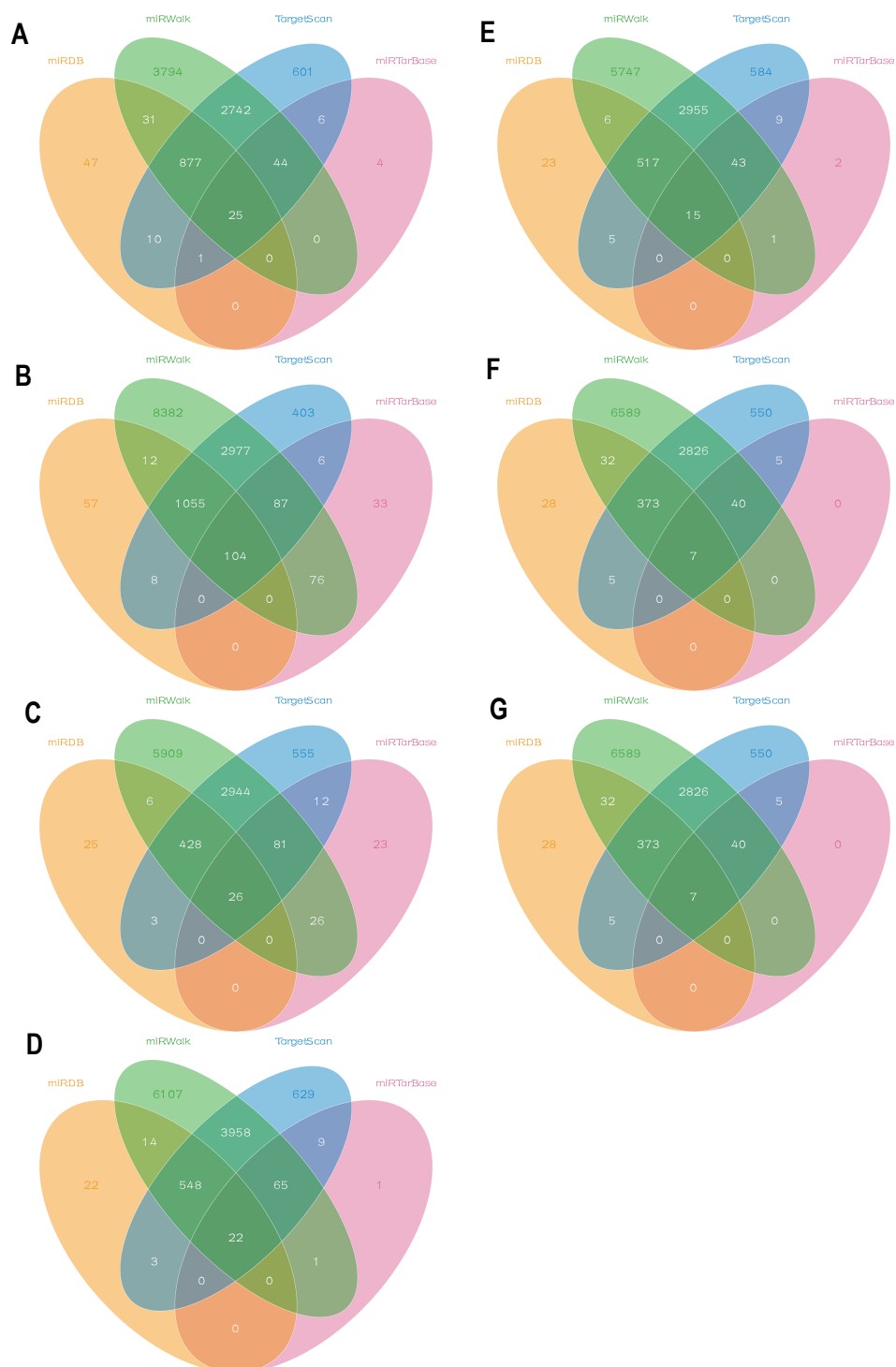

**Figure 3  Bioinformatics analysis of selected miRNAs.** (A–G) Based on TargetScan, miRDB, miRTar-Base and miRWalk, the predict targets gene of miR-9-3p (A), miR-9-5p (B), miR-146a-5p (C), miR-370-3p (D), miR-433-3p (E), miR-519a-5p (F) and miR-522-5p (G) were selected using Venn graphing.

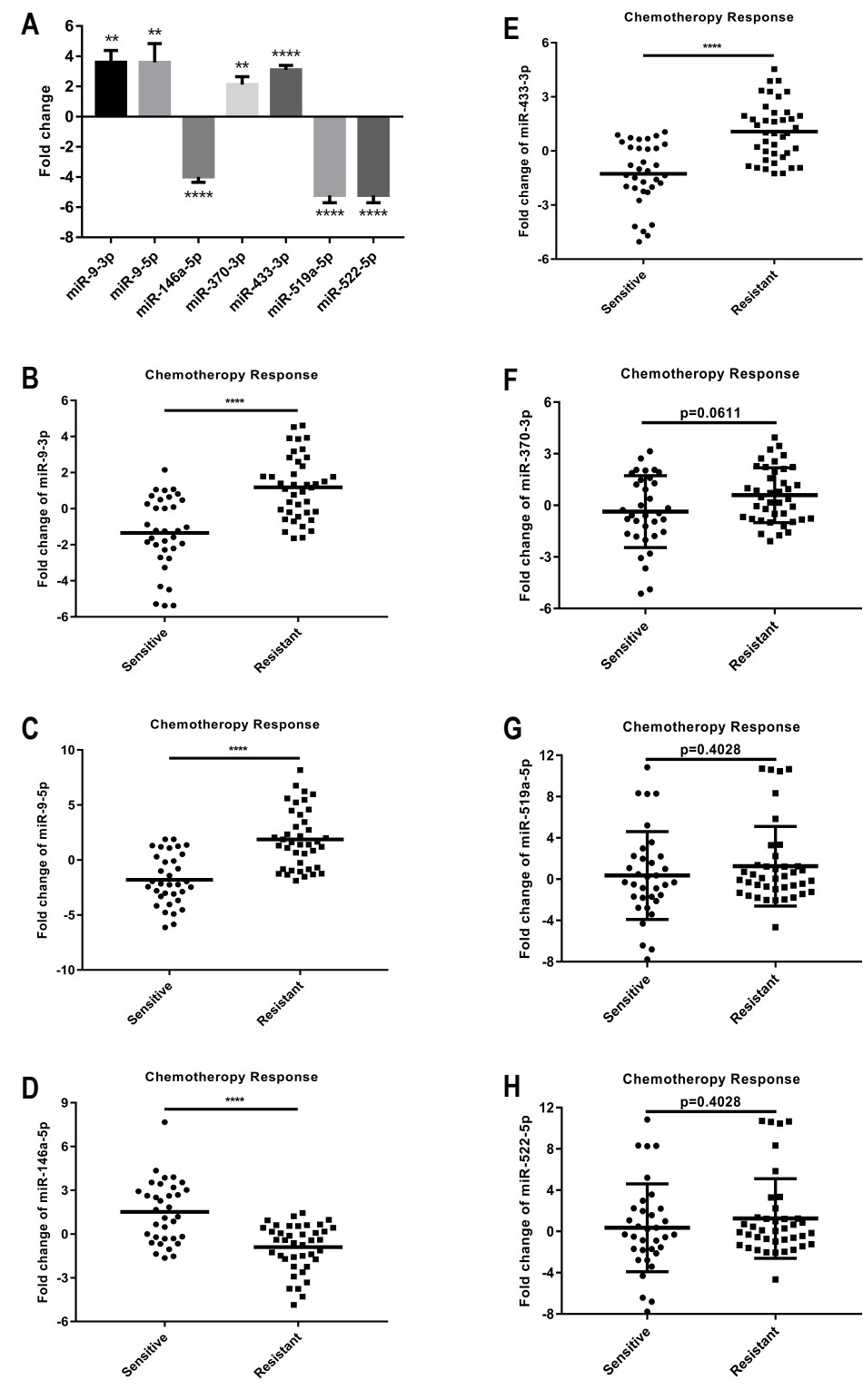

**Figure 4  Expression levels and functions of seven selected miRNAs in human GC cells and clinical specimens.** (A) The relative level (fold change) of these seven selected miRNAs between DDP-resistant

**Figure 4 (…continued)**
MGC803/DDP cells and parental MGC803 cells was analyzed via RT-qPCR. (B–E) The relative levels of miR-9-3p (B), miR-9-5p (C), miR-146a-5p (D), and miR-433-3p (E) between 34 chemotherapy response sensitive gastric cancer serums and 40 chemotherapy response resistant gastric cancer serums were significantly different, which were measured using RT-qPCR. (F–H) The relative levels of miR-370-3p (F), miR-519a-5p (G), and miR-522-5p (H) between 34 chemotherapy response sensitive gastric cancer serums and 40 chemotherapy response resistant gastric cancer serums didn't show significantly different, which were measured using RT-qPCR. Each assay was conducted in triplicate. ****$P < 0.0001$, ***$P < 0.001$, **$P < 0.01$, *$P < 0.05$ and mean $\pm$ SD were utilized to show the data.

**Table 3  7 selected miRNAs in sRNA-seq data and RT-qPCR verification in GC cells.**

| miRNA | sRNA-seq | | RT-qPCR | |
|---|---|---|---|---|
| | log$_2$FC | P-value | log$_2$FC | P-value |
| hsa-miR-9-3p | −7.5647 | 0.0002 | 3.6005 | 0.0013** |
| hsa-miR-9-5p | −7.2625 | 1.0889E-15 | 3.5865 | 0.0076** |
| hsa-miR-146a-5p | 6.1991 | 4.1414E-11 | −3.9956 | <0.0001**** |
| hsa-miR-370-3p | −6.8591 | 0.0029 | 2.1172 | 0.0022** |
| hsa-miR-433-3p | −6.4043 | 0.0092 | 3.1024 | <0.0001**** |
| hsa-miR-519a-5p | −6.5723 | 0.0061 | −5.2120 | <0.0001**** |
| hsa-miR-522-5p | −6.5723 | 0.0061 | −5.2120 | <0.0001**** |

The clinical parameter findings of 74 patients with neoadjuvant chemotherapy or palliative treatment are shown in Table 4. Using the median ratio as the cutoff for the relative miRNA expression (7 miRNA candidates; Fold Change = 0) in serum, patients were classified into two groups: low miRNA (7 miRNA candidates) and high miRNA (7 miRNA candidates). Clinical data was collected and efficacy was evaluated by CT in 2–3 cycles following chemotherapy. Independent measurements were made by two radiologists according to CHOI's principle (Choi et al., 2004). Unenhanced-combined-enhanced CT was applied to accurately evaluate the tumor response using tumor density measurement. RT-qPCR analysis was used to detect miRNA (7 miRNA candidates) levels in 34 chemotherapy response-sensitive (CR and PR) and 40 chemotherapy response-resistant (SD and PD) patients to validate miRNA (7 miRNA candidates) expression levels.

Our results proved that miR-9-3p expression obviously increased in the serum samples of chemotherapy response-resistant GC patients; 67.5% (27 of 40) of the high miR-9-3p samples showed DDP resistance ($p < 0.0001$, Fig. 4B). miR-9-5p expression increased in the serum samples of chemotherapy response-resistant GC patients; 72.5% (29 of 40) of the high miR-9-5p samples showed DDP resistance ($p < 0.0001$, Fig. 4C). miR-146a-5p expression was obviously induced in the serum samples of chemotherapy response-resistant GC patients; 62.5% (25 of 40) of the low miR-146a-5p samples showed DDP resistance ($p < 0.0001$, Fig. 4D). miR-433-3p expression increased in the serum samples of chemotherapy response-resistant GC patients; 67.5% (27 of 40) of the high miR-433-3p samples showed DDP resistance ($p < 0.0001$, Fig. 4E). However, there was no significant difference between the relative levels of miR-370-3p ($p =0.0611$, Fig. 4F), miR-519a-5p ($p = 0.4028$, Fig. 4G), and miR-522-5p ($p = 0.4028$, Fig. 4H) in the serum

**Table 4  The correlation between miRNA expression and clinical parameters in GC patients ($n = 74$).**

| miRNA level | Cases (n) | Gender | | | Age (years) | | | Chemotherapy response | | |
|---|---|---|---|---|---|---|---|---|---|---|
| | | Male | Female | *P*-value | <60 | ≥60 | *P*-value | Sensitive | Resistant | *P*-value |
| **miR-9-3p** | 74 | 52 | 22 | | 23 | 51 | | 34 | 40 | |
| low | 37 | 24 | 13 | 0.2413 | 9 | 28 | 0.5856 | 24 | 13 | <0.0001**** |
| high | 37 | 28 | 9 | | 14 | 23 | | 10 | 27 | |
| **miR-9-5p** | 74 | 52 | 22 | | 23 | 51 | | 34 | 40 | |
| low | 37 | 28 | 9 | 0.8557 | 11 | 26 | 0.9262 | 26 | 11 | <0.0001**** |
| high | 37 | 24 | 13 | | 12 | 25 | | 8 | 29 | |
| **miR-146a-5p** | 74 | 52 | 22 | | 23 | 51 | | 34 | 40 | |
| low | 37 | 26 | 11 | 0.7469 | 12 | 25 | 0.6510 | 12 | 25 | <0.0001**** |
| high | 37 | 26 | 11 | | 11 | 26 | | 22 | 15 | |
| **miR-370-3p** | 74 | 52 | 22 | | 23 | 51 | | 34 | 40 | |
| low | 37 | 23 | 14 | 0.1757 | 13 | 24 | 0.8257 | 20 | 17 | 0.0611 |
| high | 37 | 29 | 8 | | 10 | 27 | | 14 | 23 | |
| **miR-433-3p** | 74 | 52 | 22 | | 23 | 51 | | 34 | 40 | |
| low | 37 | 24 | 13 | 0.1848 | 9 | 28 | 0.3408 | 24 | 13 | <0.0001**** |
| high | 37 | 28 | 9 | | 14 | 23 | | 10 | 27 | |
| **miR-519a-5p** | 74 | 52 | 22 | | 23 | 51 | | 34 | 40 | |
| low | 37 | 24 | 13 | 0.4414 | 9 | 28 | 0.1552 | 18 | 19 | 0.4028 |
| high | 37 | 28 | 9 | | 14 | 23 | | 16 | 21 | |
| **miR-522-5p** | 74 | 52 | 22 | | 23 | 51 | | 34 | 40 | |
| low | 37 | 24 | 13 | 0.4414 | 9 | 28 | 0.1552 | 18 | 19 | 0.4028 |
| high | 37 | 28 | 9 | | 14 | 23 | | 16 | 21 | |

of 34 chemotherapy response-sensitive gastric cancer samples and 40 chemotherapy response-resistant gastric cancer samples.

Our results indicate that miR-9-3p, miR-9-5p, miR-146a-5p, and miR-433-3p may act as potential new biomarkers for the chemotherapy response of DDP treatment. The levels of miR-9-3p, miR-9-5p, miR-146a-5p, and miR-433-3p were not related to gender and age (Table 4).

### Receiver operating characteristic (ROC) analysis of miRNAs

Four miRNAs distinguished the GC chemotherapy response-resistant (SD+PD) group from the GC chemotherapy response-sensitive group (CR+PR). The results for the area under the curves (AUC), standard deviation (SD), 95% confidence intervals (CI), *P*-values, sensitivities (SE), and specificities (SP) of these miRNAs were as follows (Table 5): miR-9-3p (AUC = 0.824, Fig. 5A), miR-9-5p (AUC = 0.856, Fig. 5B), miR-146a-5p (AUC = 0.799, Fig. 5C) and miR-433-3p (AUC = 0.838, Fig. 5D), respectively. These four miRNAs provided promising AUC values for differentiating the GC chemotherapy response-resistant groups from the GC chemotherapy response-sensitive groups.

Two of the four candidate miRNAs were combined in a logistic model, with a significantly improved performance compared with the individual miRNA, determined by (Table 5): miR-9-3p + miR-9-5p (AUC = 0.889, Fig. 6A), and the risk score factors (RSF) were

**Table 5   ROC analysis of miRNAs.**

| miRNA | AUC | SD | 95%CI | SE (%) | SP (%) | *P* value |
|---|---|---|---|---|---|---|
| miR-9-3p | 0.824 | 0.047 | 0.731–0.916 | 70.6 | 67.5 | <0.0001**** |
| miR-9-5p | 0.856 | 0.042 | 0.773–0.939 | 76.5 | 72.5 | <0.0001**** |
| miR-146a-5p | 0.799 | 0.052 | 0.697–0.900 | 64.7 | 62.5 | <0.0001**** |
| miR-433-3p | 0.838 | 0.045 | 0.750–0.925 | 70.6 | 67.5 | <0.0001**** |
| miR-9-3p + miR-9-5p | 0.889 | 0.036 | 0.818–0.960 | 82.5 | 76.5 | <0.0001**** |
| miR-9-3p + miR-146a-5p | 0.903 | 0.034 | 0.836–0.969 | 80 | 76.5 | <0.0001**** |
| miR-9-3p + miR-433-3p | 0.865 | 0.04 | 0.787–0.944 | 77.5 | 73.5 | <0.0001**** |
| miR-9-5p + miR-146a-5p | 0.901 | 0.034 | 0.834–0.967 | 80 | 79.4 | <0.0001**** |
| miR-9-5p + miR-433-3p | 0.898 | 0.034 | 0.830–0.965 | 77.5 | 76.5 | <0.0001**** |
| miR-146a-5p + miR-433-3p | 0.885 | 0.037 | 0.813–0.956 | 77.5 | 73.5 | <0.0001**** |
| miR-9-3p + miR-9-5p + miR-146a-5p | 0.93 | 0.027 | 0.877–0.983 | 82.5 | 73.5 | <0.0001**** |
| miR-9-3p + miR-9-5p + miR-433-3p | 0.915 | 0.03 | 0.856–0.975 | 80 | 79.4 | <0.0001**** |
| miR-9-3p + miR-146a-5p + miR-433-3p | 0.918 | 0.031 | 0.858–0.979 | 82.5 | 82.4 | <0.0001**** |
| miR-9-5p + miR-146a-5p + miR-433-3p | 0.926 | 0.028 | 0.871–0.980 | 85 | 76.5 | <0.0001**** |
| miR-9-3p + miR-9-5p + miR-146a-5p + miR-433-3p | 0.937 | 0.025 | 0.887–0.987 | 82.5 | 76.5 | <0.0001**** |

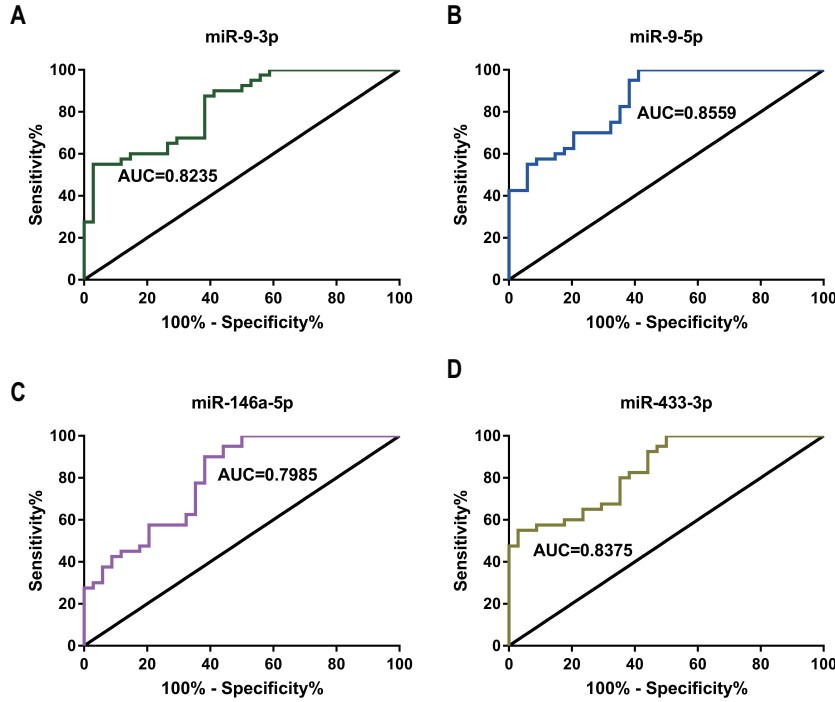

**Figure 5   ROC curve and AUC value in comparison of the prognostic accuracy for DDP response with single miRNA expression.** (A–D) ROC curves and AUC values of miR-9-3p (A), miR-9-5p (B), miR-146a-5p (C) and miR-433-3p (D) distinguished the GC chemotherapy response-resistant group from the GC chemotherapy response-sensitive group.

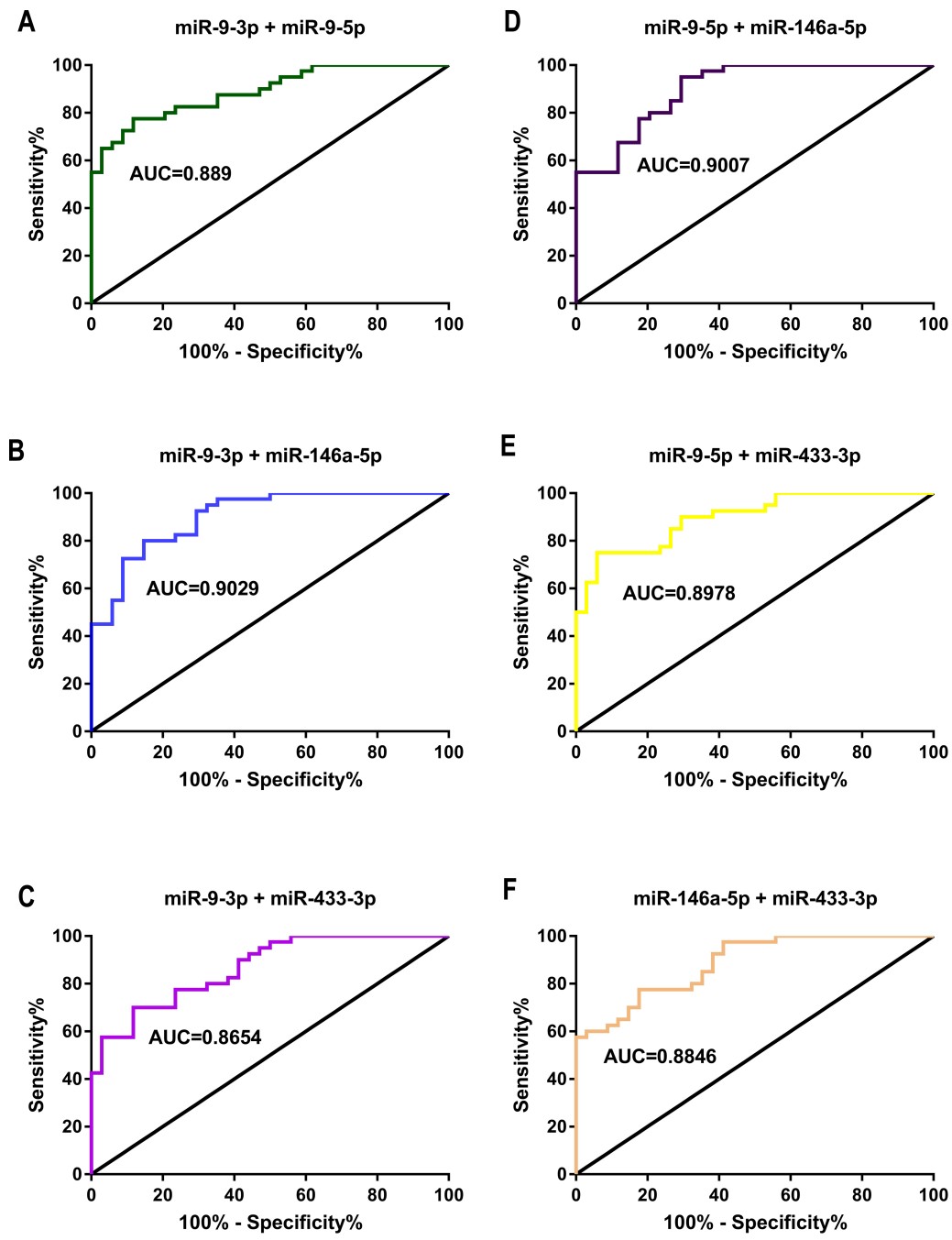

**Figure 6** **ROC curve and AUC value in comparison of the prognostic accuracy for DDP response with two combined miRNAs expression.** (A–F) ROC curves and AUC values of miR-9-3p combined with miR-9-5p (A), miR-9-3p combined with miR-146a-5p (B), miR-9-3p combined with miR-433-3p (C), miR-9-5p combined with miR-146a-5p (D), miR-9-5p combined with miR-433-3p (E) and miR-146a-5p combined with miR-433-3p (F) distinguished the GC chemotherapy response-resistant group from the GC chemotherapy response-sensitive group.

calculated as $0.731 \times$ miR-9-3p $+ 0.586 \times$ miR-9-5p $+ 0.192$; miR-9-3p $+$ miR-146a-5p (AUC $= 0.903$, Fig. 6B), and RSF $= 0.800 \times$ miR-9-3p-0.772 $\times$ miR-146a-5p $+ 0.392$; miR-9-3p $+$ miR-433-3p (AUC $= 0.865$, Fig. 6C), and RSF $= 0.574 \times$ miR-9-3p $+ 0.684 \times$ miR-433-3p $+ 0.173$; miR-9-5p $+$ miR-146a-5p (AUC $= 0.901$, Fig. 6D), and RSF $= 0.575 \times$ miR-9-5p $- 0.746 \times$ miR-146a-5p $+ 0.345$; miR-9-5p $+$ miR-433-3p (AUC $= 0.898$, Fig. 6E), and RSF $= 0.551 \times$ miR-9-5p $+ 0.847 \times$ miR-433-3p $+ 0.264$; miR-146a-5p $+$ miR-433-3p (AUC $= 0.885$, Fig. 6F), and RSF $= -0.637 \times$ miR-146a-5p $+ 0.802 \times$ miR-433-3p $+ 0.327$.

Three or more of the four candidate miRNAs were combined resulting in a significantly improved performance compared with the individual miRNA (Table 5: miR-9-3p $+$ miR-9-5p $+$ miR-146a-5p) (AUC $= 0.930$, Fig. 7A), and the RSF was calculated as $0.690 \times$ miR-9-3p $+ 0.515 \times$ miR-9-5p $- 0.738 \times$ miR-146a-5p $+ 0.386$; miR-9-3p $+$ miR-9-5p $+$ miR-433-3p (AUC $= 0.915$, Fig. 7B), RSF $= 0.514 \times$ miR-9-3p $+ 0.526 \times$ miR-9-5p $+ 0.614 \times$ miR-433-3p $+ 0.240$; miR-9-3p $+$ miR-146a-5p $+$ miR-433-3p (AUC $= 0.918$, Fig. 7C), RSF $= 0.646 \times$ miR-9-3p $- 0.700 \times$ miR-146a-5p $+ 0.504 \times$ miR-433-3p $+ 0.388$; miR-9-5p $+$ miR-146a-5p $+$ miR-433-5p (AUC $= 0.926$, Fig. 7D), RSF $= 0.521 \times$ miR-9-5p $- 0.650 \times$ miR-146a-5p $+ 0.655 \times$ miR-433-3p $+ 0.376$; miR-9-3p $+$ miR-9-5p $+$ miR-146a-5p $+$ miR-433-3p (AUC $= 0.937$, Fig. 7E), RSF $= 0.540 \times$ miR-9-3p $+ 0.480 \times$ miR-9-5p $- 0.681 \times$ miR-146a-5p $+ 0.393 \times$ miR-433-3p $+ 0.398$.

## Pairwise comparison of ROC curves

MedCalc software was used to display the data for the different ROC curves and the results of pairwise comparison of all ROC curves (difference between the areas (DBA), standard error (SE), 95% confidence interval for the difference (CI), and $P$-value) (Table 6).

The ROC curve of miR-9-5p had the best AUC value of the four miRNA candidates. A comparison between the combined group (miR-9-5p $+$ miR-9-3p $+$ miR-146a-5p), and the miR-9-5p group showed that the two compared areas were significantly different ($p = 0.026$, Fig. 8A). The compared areas of the combined group (miR-9-5p $+$ miR-9-3p $+$ miR-433-3p) and the miR-9-5p group were significantly different ($p = 0.045$, Fig. 8B). The compared areas of the combined group (miR-9-5p $+$ miR-146a-5p $+$ miR-433-3p) and the miR-9-5p group were significantly different ($p = 0.034$, Fig. 8C) and the compared areas of the combined group (miR-9-5p $+$ miR-9-3p $+$ miR-146a-5p $+$ miR-433-3p) and the miR-9-5p group were also significantly different ($p = 0.016$, Fig. 8D). However, the compared areas of the other combined group and the miR-9-5p group were not significantly different ($P > 0.05$).

## Validate selected miRNAs in public database

To determine the role of miR-9-3p, miR-9-5p, miR-146a-5p, miR-433-3p, or their combination as a potential prognostic factor, a large cohort analysis was conducted using Kaplan–Meier survival data according to the Kaplan–Meier Plotter database, the OncoLnc database, and the OncomiR database (Table 7).

We observed that patients with higher miR-9-3p levels had shorter OS times than those with lower miR-9-3p levels in the OncoLnc database ($p = 0.00836$, Fig. 9A) and the

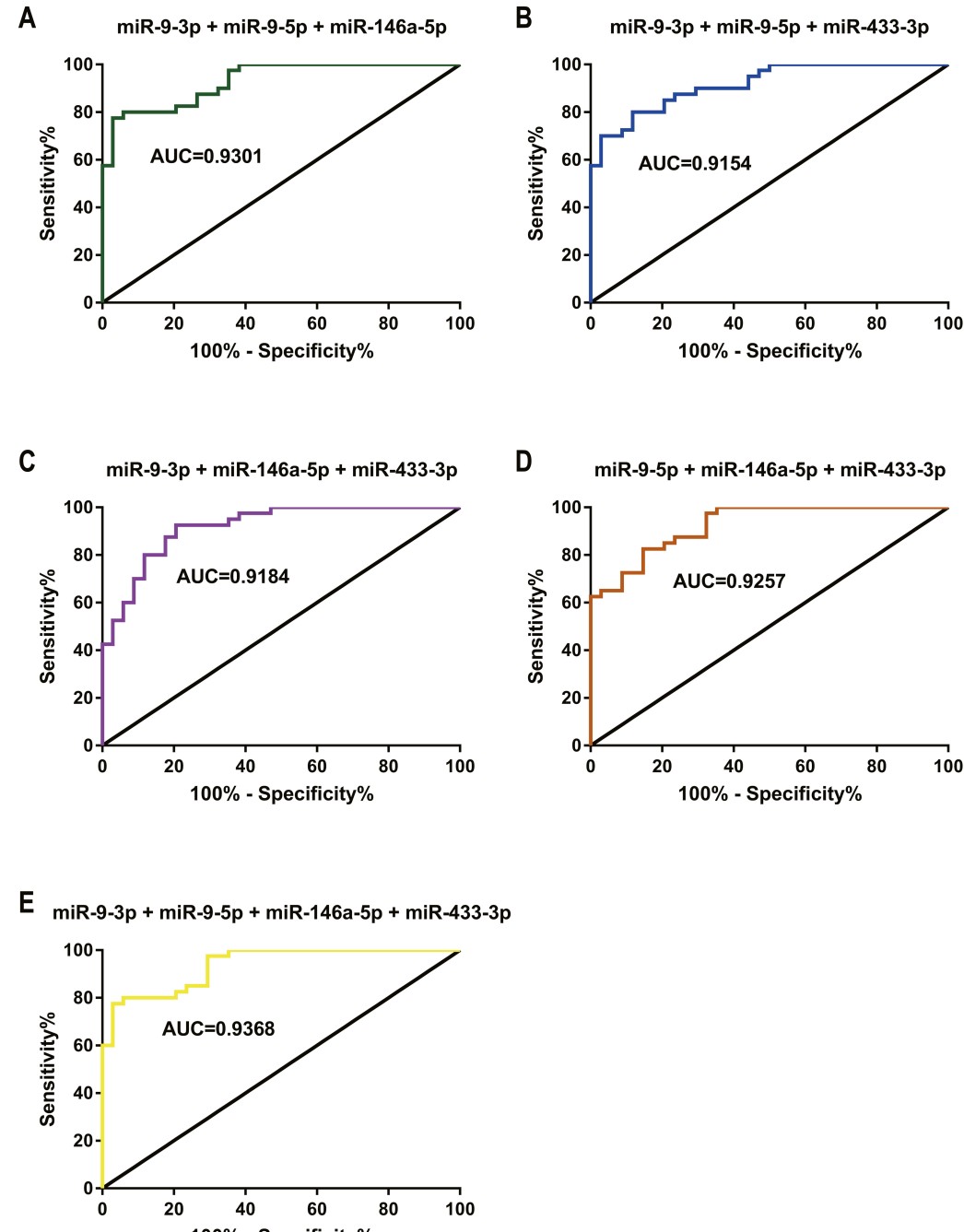

**Figure 7** **ROC curve and AUC value in comparison of the prognostic accuracy for DDP response with three or more combined miRNAs expression.** (A–E) ROC curves and AUC values of miR-9-3p combined with miR-9-5p and miR-146a-5p (A), miR-9-3p combined with miR-9-5p and miR-433-3p (B), miR-9-3p combined with miR-146a-5p and miR-433-3p (C), miR-9-5p combined with miR-146a-5p and miR-433-3p (D), and miR-9-3p combined with miR-9-5p, miR-146a-5p and miR-433-3p (E) distinguished the GC chemotherapy response-resistant group from the GC chemotherapy response-sensitive group.

**Table 6  Pairwise comparison of ROC curves.**

| Compared areas | DBA | SE | 95%CI | *P* value |
|---|---|---|---|---|
| miR-9-5p versus miR-9-5p + miR-9-3p | 0.034 | 0.027 | −0.019∼0.088 | 0.212 |
| miR-9-5p versus miR-9-5p + miR-146a-5p | 0.046 | 0.034 | −0.021∼0.113 | 0.176 |
| miR-9-5p versus miR-9-5p + miR-433-3p | 0.043 | 0.028 | −0.012∼0.098 | 0.127 |
| miR-9-5p + miR-9-3p versus miR-9-5p + miR-9-3p + miR-146a-5p | 0.041 | 0.025 | −0.007∼0.089 | 0.094 |
| miR-9-5p + miR-9-3p versus miR-9-5p + miR-9-3p + miR-433-3p | 0.026 | 0.018 | −0.009∼0.062 | 0.143 |
| miR-9-5p + miR-146a-5p versus miR-9-5p + miR-146a-5p + miR-9-3p | 0.029 | 0.019 | −0.009∼0.068 | 0.131 |
| miR-9-5p + miR-146a-5p versus miR-9-5p + miR-146a-5p + miR-433-3p | 0.015 | 0.033 | −0.050∼0.079 | 0.654 |
| miR-9-5p + miR-433-3p versus miR-9-5p + miR-433-3p + miR-9-3p | 0.018 | 0.018 | −0.017∼0.052 | 0.314 |
| miR-9-5p + miR-433-3p versus miR-9-5p + miR-433-3p + miR-146a-5p | 0.028 | 0.022 | −0.016∼0.072 | 0.21 |
| miR-9-5p + miR-9-3p + miR-146a-5p versus miR-9-5p + miR-9-3p + miR-146a-5p + miR-433-3p | 0.007 | 0.01 | −0.013∼0.026 | 0.514 |
| miR-9-5p + miR-9-3p + miR-433-3p versus miR-9-5p + miR-9-3p + miR-433-3p + miR-146a-5p | 0.021 | 0.02 | −0.018∼0.061 | 0.288 |
| miR-9-5p + miR-146a-5p + miR-433-3p versus miR-9-5p + miR-146a-5p + miR-433-3p + miR-9-3p | 0.011 | 0.013 | −0.013∼0.036 | 0.378 |
| miR-9-5p versus miR-9-5p + miR-9-3p + miR-146a-5p | 0.075 | 0.034 | 0.009∼0.142 | 0.026* |
| miR-9-5p versus miR-9-5p + miR-9-3p + miR-433-3p | 0.061 | 0.03 | 0.001∼0.120 | 0.045* |
| miR-9-5p versus miR-9-5p + miR-146a-5p + miR-433-3p | 0.071 | 0.034 | 0.005∼0.137 | 0.034* |
| miR-9-5p versus miR-9-5p + miR-9-3p + miR-146a-5p + miR-433-3p | 0.082 | 0.034 | 0.015∼0.149 | 0.016* |
| miR-9-5p + miR-9-3p versus miR-9-5p + miR-9-3p + miR-146a-5p + miR-433-3p | 0.048 | 0.026 | −0.003∼0.099 | 0.066 |
| miR-9-5p + miR-146a-5p versus miR-9-5p + miR-146a-5p + miR-9-3p + miR-433-3p | 0.036 | 0.021 | −0.004∼0.077 | 0.081 |
| miR-9-5p + miR-433-3p versus miR-9-5p + miR-433-3p + miR-9-3p + miR-146a-5p | 0.039 | 0.025 | −0.011∼0.089 | 0.124 |

OncomiR database ($p = 0.06643$, Fig. 9B). Patients with higher miR-9-5p levels had shorter OS times than those with lower miR-9-5p levels in the OncoLnc database ($p = 0.00181$, Fig. 9C) and the OncomiR database ($p = 0.0044$, Fig. 9D). Patients with higher miR-433-3p levels had shorter OS times than those with lower miR-433-3p levels in the OncoLnc database ($p = 0.0311$, Fig. 9E) and the OncomiR database ($p = 0.05651$, Fig. 9F). These observations indicate that the high expression of miR-9-3p, miR-9-5p, or miR-433-3p are a valid biomarker for chemoresistance and poor survival, especially miR-9-5p.

Patients with higher levels of the combined group (miR-9-5p + miR-9-3p) had shorter OS times than those with lower combined levels in the OncomiR database ($p = 0.003421$, Fig. 10A). Patients with higher levels of the combined group (miR-9-5p + miR-433-3p) had shorter OS times than those with lower combined levels in the OncomiR database ($p = 0.01428$, Fig. 10B). Patients with higher levels of the combined group (miR-9-5p +

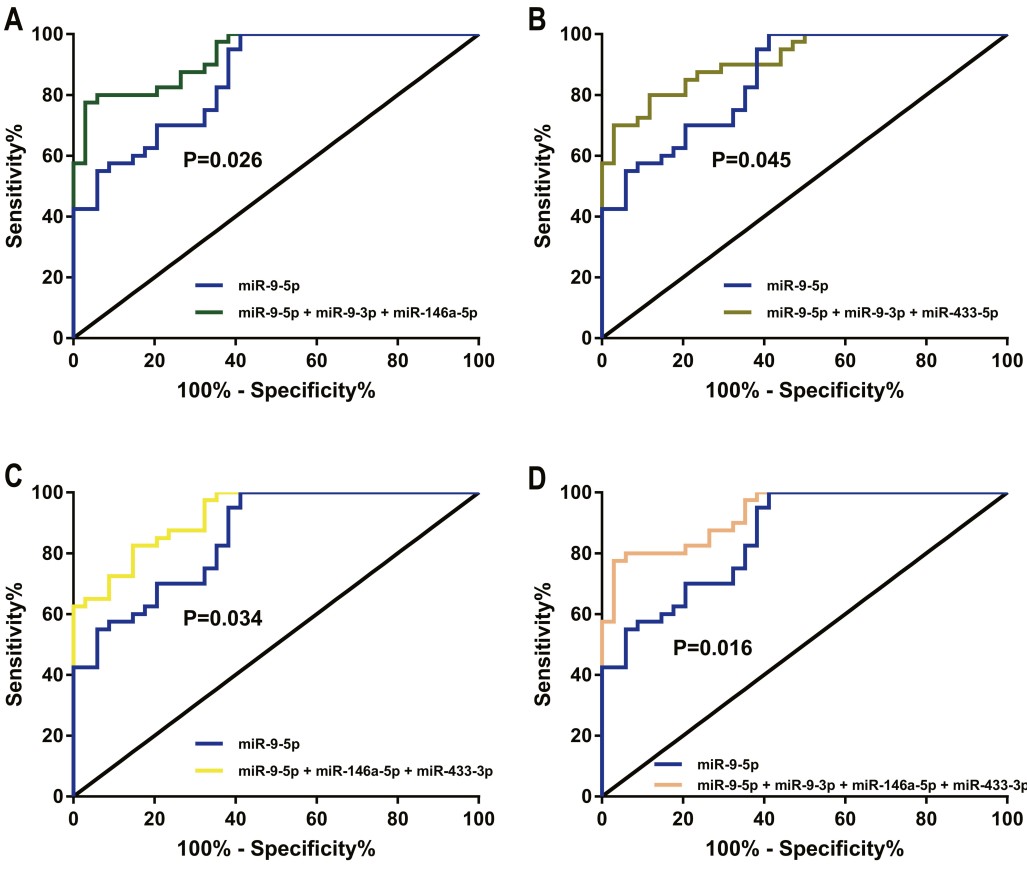

**Figure 8** **Pairwise comparison of ROC curves by the MedCalc software.** (A) The two compared areas between the miR-9-5p + miR-9-3p + miR-146a-5p group and the miR-9-5p group were significantly different. (B)The two compared areas between the miR-9-5p + miR-9-3p + miR-433-3p group and the miR-9-5p group were significantly different. (C) The two compared areas between the miR-9-5p + miR-146a-5p + miR-433-3p group and the miR-9-5p group were significantly different. (D) The two compared areas between the miR-9-5p + miR-9-3p + miR-146a-5p + miR-433-3p group and the miR-9-5p group were significantly different.

miR-9-3p + miR-433-3p) had shorter OS times than those with lower combined levels in the OncomiR database ($p = 0.008066$, Fig. 10C). Patients with higher levels of the combined group (miR-9 + miR-433) had shorter OS times than those with lower combined levels in the Kaplan–Meier Plotter database ($p = 0.0047$, Fig. 10D). These observations indicate that higher levels of the combined groups, especially the combined group (miR-9-5p + miR-9-3p + miR-433-3p) are a valid biomarker for chemoresistance and poor survival.

## DISCUSSION

DDP resistance is a barrier to the effective treatment of GC. The identification of unique biomarkers for drug resistance can help to accurately diagnose and treat GC.

The involvement of non-coding nucleic acids, such as miRNAs in response to DDP treatment, is not well understood. miRNAs play vital roles in the progression and development of tumors, and may be correlated with resistance to chemotherapeutics

**Table 7  Four selected miRNAs and their combinations in three public databases for prognostic analysis.**

| miRNA and Combination | OncoLnc | OncomiR | Kaplan–Meier plotter |
|---|---|---|---|
| | *P*-value | *P*-value | *P*-value |
| miR-9-3p | 0.0084** | 0.0664 | 0.0047** |
| miR-9-5p | 0.0018** | 0.0044** | 0.0047** |
| miR-146a-5p | 0.1460 | 0.0460* | 0.0230* |
| miR-433-3p | 0.0311* | 0.0565 | 0.0017** |
| miR-9-3p + miR-9-5p | None | 0.0034** | 0.0047** |
| miR-9-3p + miR-146a-5p | None | 0.6535 | 0.0017** |
| miR-9-3p + miR-433-3p | None | 0.2405 | 0.0047** |
| miR-9-5p + miR-146a-5p | None | 0.3989 | 0.0017** |
| miR-9-5p + miR-433-3p | None | 0.0143* | 0.0047** |
| miR-146a-5p + miR-433-3p | None | 0.1278 | 0.0230* |
| miR-9-3p + miR-9-5p + miR-146a-5p | None | 0.6655 | 0.0017** |
| miR-9-3p + miR-9-5p + miR-433-3p | None | 0.0081** | 0.0047** |
| miR-9-3p + miR-146a-5p + miR-433-3p | None | 0.5808 | 0.0020** |
| miR-9-5p + miR-146a-5p + miR-433-3p | None | 0.5782 | 0.0020** |
| miR-9-3p + miR-9-5p + miR-146a-5p + miR-433-3p | None | 0.4551 | 0.0020** |

in tumor cells (*Xia et al., 2008*; *Yang et al., 2008*; *Li et al., 2009*). The downregulation of miR-21 has been shown to alter survival rates by increasing DDP sensitivity in GC cells (*Yang et al., 2013*).

Liquid biopsies can identify markers from blood and various body fluids and are minimally invasive, safe, economical, and convenient versus the use of more invasive tests or biopsies for the diagnosis of GC (*Tsujiura et al., 2014*). Liquid biopsies are suitable for screening a wide range of people, which can improve the diagnosis and treatment of GC. Several studies have been conducted on the use of miRNAs in GC patients related to the occurrence, development, diagnosis, treatment, and prognosis of the disease. *Fang et al. (2013)* reported that certain carcinogenesis-related miRNAs (miR-10b, miR-21, miR-223, and miR-338) and tumor suppressor miRNAs (miR-30a-5p, miR-126, and let-7a) can be used as prognosis markers in GC patients. Numerous studies have reported on the use of combinations of circulating miRNAs for greater diagnostic accuracy, which is indicated by an area under the ROC curve larger than 0.8 (*Fang et al., 2013*; *Ng et al., 2009*).

Relatively few studies have reported the application of miRNAs in the diagnosis and screening of the chemotherapy response of GC. In this study, the differences in the expression profiles of miRNAs in chemotherapy response-sensitive and chemotherapy response-resistant GC cells were established using sRNA-seq and the potential miRNAs were screened using bioinformatics analyses.

Our results showed that miR-9-3p, miR-9-5p, and miR-433-3p were significantly up-regulated and miR-146a-5p was significantly down-regulated in the MGC803/DDP cells and in chemotherapy response-resistant GC patients. While some of these miRNAs

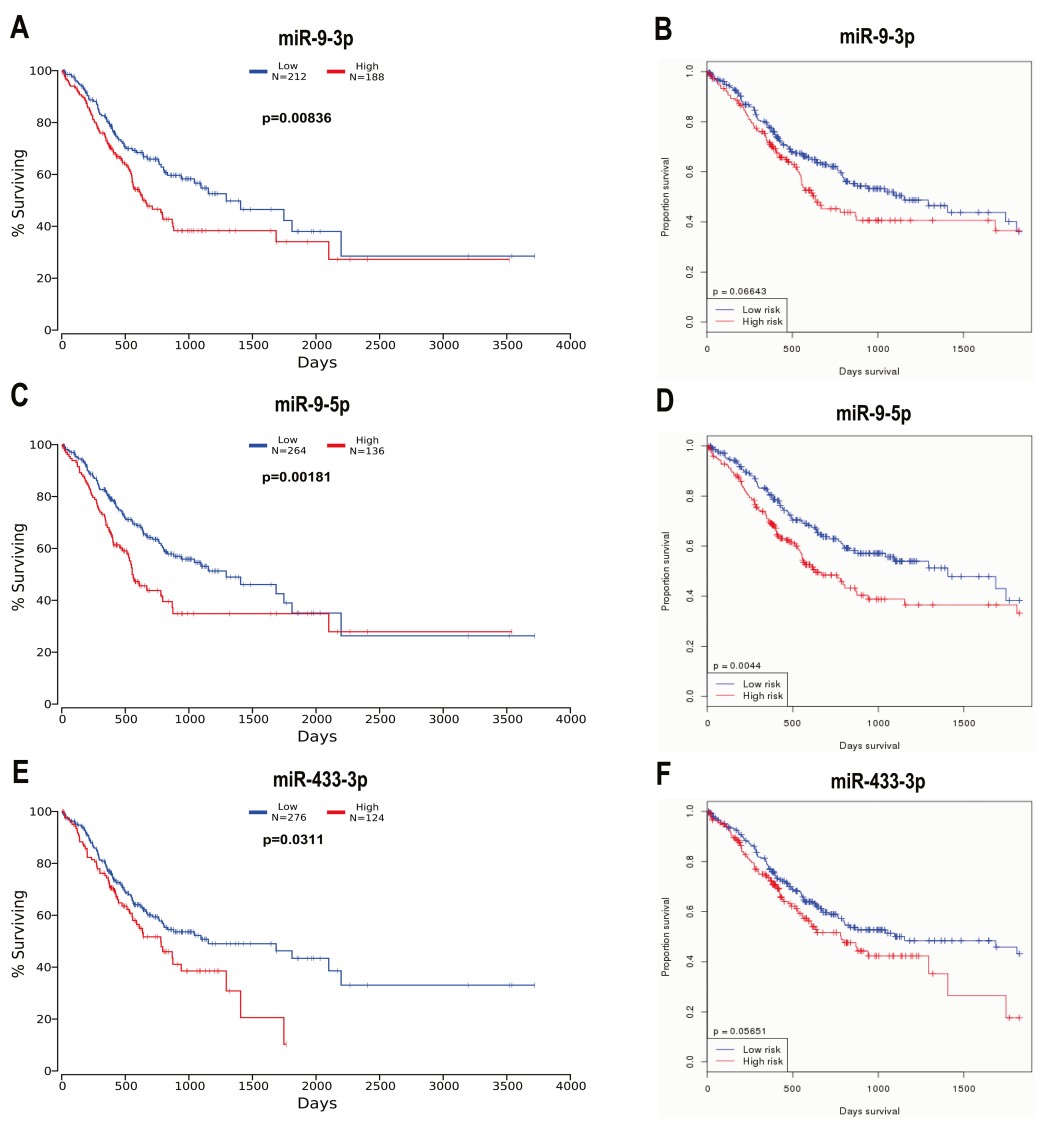

**Figure 9** **Three selected miRNAs were ascertained in two public databases for prognostic analysis.** (A, B) Kaplan-Meier survival curves suggested that patients with high miR-9-3p levels had shorter OS times than those with low miR-9-3p levels in the OncoLnc database (A, $P = 0.00836$) and in the OncomiR database (B, $P = 0.06643$). (C, D) Kaplan-Meier survival curves suggested that patients with high miR-9-5p levels had shorter OS times than those with low miR-9-5p levels in the OncoLnc database (C, $P = 0.00181$) and in the OncomiR database (D, $P = 0.0044$). (E, F) Kaplan-Meier survival curves suggested that patients with high miR-433-3p levels had shorter OS times than those with low miR-433-3p levels in the OncoLnc database (E, $P = 0.0311$) and in the OncomiR database (F, P=0.05651).

are considered to be diagnostic or prognostic biomarkers in GC, ours is the first study to explore their potential use in determining the chemotherapy response in GC patients.

Numerous studies have shown that a combination of multiple miRNAs could more easily identify early-stage CC patients than a single miRNA biomarker (*Vychytilova-Faltejskova et al., 2016*; *Ng et al., 2009*). We combined the three miRNAs in our study (miR-9-3p, miR-9-5p and miR-433-3p) to achieve an AUC of 0.915. The equation used in this

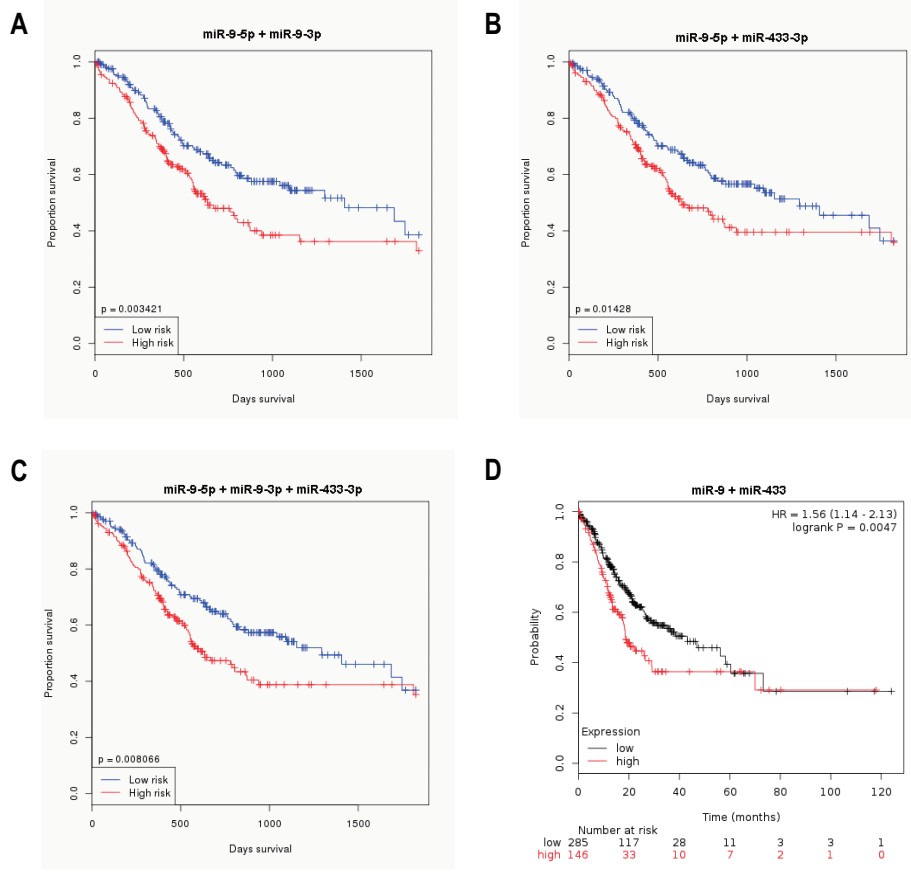

**Figure 10** **Different miRNA combinations were ascertained in two public databases for prognostic analysis.** (A) Kaplan-Meier survival curves suggested that patients with high miR-9-3p + miR-9-5p levels had shorter OS times than those with low miR-9-3p + miR-9-5p levels in the OncomiR database ($P = 0.003421$). (B) Kaplan-Meier survival curves suggested that patients with high miR-9-5p + miR-433-3p levels had shorter OS times than those with low miR-9-5p + miR-433-3p levels in the OncomiR database ($P = 0.01428$). (C) Kaplan-Meier survival curves suggested that patients with high miR-9-3p + miR-9-5p + miR-433-3p levels had shorter OS times than those with low miR-9-3p + miR-9-5p + miR-433-3p levels in the OncomiR database ($P = 0.008066$). (D) Kaplan-Meier survival curves suggested that patients with high miR-9 + miR-433 levels had shorter OS times than those with low miR-9 + miR-433 levels in the Kaplan–Meier Plotter database ($P = 0.0047$).

determination was risk score factor (RSF) = $0.514 \times$ miR-9-3p + $0.526 \times$ miR-9-5p + $0.614 \times$ miR-433-3p + $0.240$. There was a significant difference between the two compared areas of the combined group (miR-9-3p + miR-9-5p + miR-433-3p) and the miR-9-5p group ($p = 0.045$), whereas the two compared areas of the other combined group and the miR-9-5p group was not significantly different ($p > 0.05$). A large cohort analysis was conducted using Kaplan–Meier survival data according to the Kaplan–Meier Plotter database, the OncoLnc database, and the OncomiR database to determine the role of miR-9-3p, miR-9-5p, miR-433-3p or their combination as a potential prognostic factor. The results indicate that the high expression of miR-9-5p and the combined group (miR-9-5p + miR-9-3p + miR-433-3p) can be used as a valid biomarker for chemoresistance and

poor survival. Serum-derived miRNAs have the potential to be used as novel noninvasive tumor markers for the chemotherapy response in GC patients.

Previous studies have shown that miR-9-5p expression may act as a potential tumor suppressor gene and is closely related to the malignant progression of GC (*Fan et al., 2019*). miR-9-3p may play an important role in tumor invasion and have potential effects on the prognosis of gastric cancer (*Meng et al., 2017*). miR-433-3p may function as a potential diagnostic marker and therapeutic target for glioma (*Sun et al., 2017*). These miRNAs serve as biomarkers for tumors and tumor progression.

## CONCLUSIONS

In conclusion, we reported on the potential for 2 serum-based biomarkers, miR-9-5p and a combined group (miR-9-5p + miR-9-3p + miR-433-3p), to predict the therapeutic benefit of DDP for GC patients. Additional clinical samples should be collected to validate these serum biomarkers.

## ACKNOWLEDGEMENTS

We thank Biobank Resource and Clinical Data of Beijing Friendship Hospital, Capital Medical University for their collection of serum samples and their data processing services.

### Funding

Our work was supported by the National Natural Science Foundation of China (8160140024) as well as the National Key Technologies R&D Program (2015BAI13B09). The funders had no role in study design, data collection and analysis, decision to publish, or preparation of the manuscript.

### Grant Disclosures

The following grant information was disclosed by the authors:
National Natural Science Foundation of China: 8160140024.
National Key Technologies R&D Program: 2015BAI13B09.

### Competing Interests

The authors declare there are no competing interests.

### Author Contributions

- Lei Jin performed the experiments, prepared figures and/or tables, authored or reviewed drafts of the paper, and approved the final draft.
- Nan Zhang performed the experiments, analyzed the data, prepared figures and/or tables, authored or reviewed drafts of the paper, and approved the final draft.
- Qian Zhang analyzed the data, authored or reviewed drafts of the paper, and approved the final draft.

- Guoqian Ding performed the experiments, analyzed the data, authored or reviewed drafts of the paper, and approved the final draft.
- Zhenghan Yang conceived and designed the experiments, authored or reviewed drafts of the paper, and approved the final draft.
- Zhongtao Zhang conceived and designed the experiments, authored or reviewed drafts of the paper, and approved the final draft.

## Human Ethics

The following information was supplied relating to ethical approvals (i.e., approving body and any reference numbers):

The study got the approval of the Ethics Committee of Beijing Friendship Hospital, Capital Medical University (2018-P2-045-01).

## Data Availability

The sequences are available at GenBank (PRJNA615333 and SRR11427197) and Figshare: Zhang, Zhongtao (2019): rawdata.rar. figshare. Dataset. https://doi.org/10.6084/m9.figshare.9963107.v1.

## Supplemental Information

Supplemental information for this article can be found online at http://dx.doi.org/10.7717/peerj.8943#supplemental-information.

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
