# Peer review of "Serum microRNAs as potential new biomarkers for cisplatin resistance in gastric cancer patients"

_PeerJ, doi:10.7717/peerj.8943_

## Round 0.1 · original submission · Minor Revisions

Dear Dr. Jin,

Thank you for your submission to Peer J. Based on the comments from the reviewers and as per my personal evaluation of your article, I suggest you to make minor revision of the article before it gets accepted. Please consider each comment of all reviewers to revise the article.

The main concerns are

(1) Organization and flow of the article, that's need to carefully copy edited
(2) Precisely describe the results
(3) Give the details of the primers and statistical methods under Method section
(4) A better discussion section is required
(5) Please consider the comments of Reviewer-2 in describing the miRNAs
(6) Carefully copy edit as there are many typo and language issues.

Reviewer 1 ·

Basic reporting

the article should include sufficient introduction as well as discussion , without repeating the same meaning

Experimental design

methods described with sufficient details

statistical analyses are very well.so, i think this manuscript must be reviewed by professor of statistic.

Validity of the findings

the results too long, contain details and conclusion, repeating the value present in the tables.
the results must be summarize

Additional comments

* IN abstract: conclusion must be clarify.
* the last paragraph in the introduction is not the aim but it is the methods.
* lines 59-66: there are repetition around the same meaning
* MGC803 cells were cultured with DDP in a continuous stepwise fashion for 15 months (please confirm, where is the references for that ) .
* this study include only one cell line MGC803 not two as mention.
* line 85-86: mention had received DDP-based chemotherapy as86 first-line treatment; (3) had received at least 2 cycles of chemotherapy;
then mention in line 91:Prechemotherapy serum samples of patients who received neoadjuvant chemotherapy or palliative treatment were collected
there is contrast.
* no prime sequences mention.
* housekeeping gene don,t mention under statistical analysis, must be mention under methods.
* discussion: there is repetition the results, must be more clarify.

Reviewer 2 ·

Basic reporting

No comment

Experimental design

No comment

Validity of the findings

No comment

Additional comments

1- Abbreviations should be given in the first metion.
2- In Table 1, log, FC and p values should be given as 4 digits after the comma.
3- The target genes shown in Figure 3 should be given as supplementary material. For example, in Figure 3A, 25 genes, which are common to four different databases and the potential target of miR-9, should be given as supplementary tables.
4- The “Above all, miR-519a-5p and miR-522-5p changed consistently with sRNA-seq, but the others showed the opposite results. Besides, miR-519a-5p and miR-522-5p showed the same result, for they had the same primer.” part in the Result section is not fully open. This section should be written more clearly and in detail. Also why are miR-370-3p, miR-519a-5p and miR-522-5p miRNAs not shown in GC patient serum samples? The results should be shown even if they are insignificant.
5- Stars in tables are given next to P values, not need to be given in table footnote.
6- Does Table 2 refer to GC patients or GC cell lines?
7- “miR-9-3p (AUC = 0.824, SD = 0.047, 95%CI: 0.731-0.916, p<0.0001, SE 70.6%, SP 67.5%, Figure 5A), miR-9-5p (AUC = 0.856, SD = 0.042, 95%CI: 0.773-0.939, p<0.0001, SE 76.5%, SP 72.4%, Figure 5B), miR-146a-5p (AUC = 0.799, SD = 0.052, 95%CI: 0.697-0.900, p<0.0001, SE 64.7%, SP 62.5%, Figure 5C) and miR-433-3p (AUC = 0.838, SD = 0.045, 95%CI: 0.750-0.925, p<0.0001, SE 70.6%, SP 67.5%, Figure5D)” in the result section is already given in the figure and table. There is no need to be given in the text. Data not specified in figures and tables should only be given in the text.
8- In Figure 9A, C and E, the p values should be indicated in the graph.
9- Figure 10A, B, C, and D belongs to which miRNA should be indicated.

·

Basic reporting

no comment

Experimental design

no comment

Validity of the findings

no comment

Additional comments

I think this is a research which has significant meaning, I'm impressed by the significant difference of the AUC value between combined group and miR-9-5p group, hopefully it may be used as a marker of chemoresistance of GC. But there were some problems needs to be correct before it can be accepted:
1. In the abstract, I think you should list the specific AUC, P value and also the 95% CI in the "results" part.(line 33-38)
2. You have mentioned that "MicroRNAs (miRNAs) are a recently discovered..." in line 51, I don't think so, miRNAs have been discovered in the last century and have been extensively studied, so "recently" is not suitable.
3. In figure 9 and 10, I can find three types of survival plots, could you unify them?

---

## Round 0.2 · accepted · Accept

Dear Dr. Jin,

Thank you for submission of the revised manuscript. The manuscript is now recommended for publication by reviewers and its now officially accepted.

Thank you,
Best regards,
Debmalya Barh, PhD

·

Basic reporting

no comment

Experimental design

no comment

Validity of the findings

no comment

Additional comments

no comment